# Phase glides and self-organization of atomically abrupt interfaces out of stochastic disorder in α-Ga$_2$O$_3$

Alexander Azarov [1] ✉, Javier García Fernández [1], Junlei Zhao [2] ✉, Ru He[3], Ji-Hyeon Park[4], Dae-Woo Jeon[4], Øystein Prytz [1], Flyura Djurabekova [3] & Andrej Kuznetsov [1] ✉

Disorder-induced ordering and remarkably high radiation tolerance in γ-phase of gallium oxide is a recent spectacular discovery at the intersection of the fundamental physics and electronic applications. Importantly, by far, these data were collected with initial samples in form of the thermodynamically stable β-phase of this material. Here, we investigate these phenomena starting from metastable α-phase and explain radically new trend occurring in the system. We argue that in contrast to that in β-to-γ disorder-induced transitions, the O sublattice in α-phase exhibits hexagonal close-packed structure, so that to activate α-to-γ transformation significant structural rearrangements are required in both Ga and O sublattices. Moreover, consistent with theoretical predictions, α-to-γ phase transformation requires accumulation of the substantial tensile strain to initiate otherwise impossible lattice glides. Thus, we explain the experimentally observed trends in term of the combination of disorder and strain governed process. Finally, we demonstrate atomically abrupt α/γ interfaces paradoxically self-organized out of the stochastic disorder.

Recently, gallium oxide (Ga$_2$O$_3$) has attracted the attention of a broad audience spreading from those dealing with fundamentals of the phase transitions[1–5] to device application experts[6–10]. Among the rest of the highlights, there was a discovery of disorder-induced ordering in Ga$_2$O$_3$[11–14] and high radiation tolerance of the formed structures[15]. Specifically, it was shown that even though its thermodynamically stable monoclinic polymorph (β-Ga$_2$O$_3$) could be swiftly disordered, it did not amorphized under irradiation but converted to a cubic defective spinel polymorph (γ-Ga$_2$O$_3$), remaining crystalline independently of subsequent irradiation[15]. Moreover, electronic radiation tolerance tests performed by comparing Schottky diodes fabricated out of β- and γ-polymorphs showed that the γ-Ga$_2$O$_3$-based diodes remained functional, while β-Ga$_2$O$_3$-based diodes lost their

rectification under identical irradiation conditions[16]. As explained recently, the rationale behind this remarkable β-to-γ Ga$_2$O$_3$ polymorph transformation is because the oxygen sublattice in these polymorphs, exhibiting face-centered cubic (fcc) structure, as shown in Fig. 1a, demonstrates strong recrystallization trends, while the Ga sublattice is susceptible to disorder[17,18]. Very recently, this idea was exploited to demonstrate multiple γ/β polymorph repetitions by adjusting spatial distributions of the disorder levels as a function of the irradiation temperature and ion flux[19]; as such, demonstrating "polymorph heterostructures" not being realized by conventional growth methods otherwise.

Meanwhile, understanding of the radiation phenomena in other Ga$_2$O$_3$ polymorphs is much less mature. For instance, for the

[1]University of Oslo, Department of Physics, Centre for Materials Science and Nanotechnology, PO Box 1048 Blindern, N-0316, Oslo, Norway. [2]Department of Electronic and Electrical Engineering, Southern University of Science and Technology, 518055 Shenzhen, China. [3]Department of Physics and Helsinki Institute of Physics, University of Helsinki, P.O. Box 43, FI-00014, Helsinki, Finland. [4]Korea Institute of Ceramic Engineering & Technology, Jinju, South Korea. ✉e-mail: alexander.azarov@smn.uio.no; zhaojl@sustech.edu.cn; andrej.kuznetsov@fys.uio.no

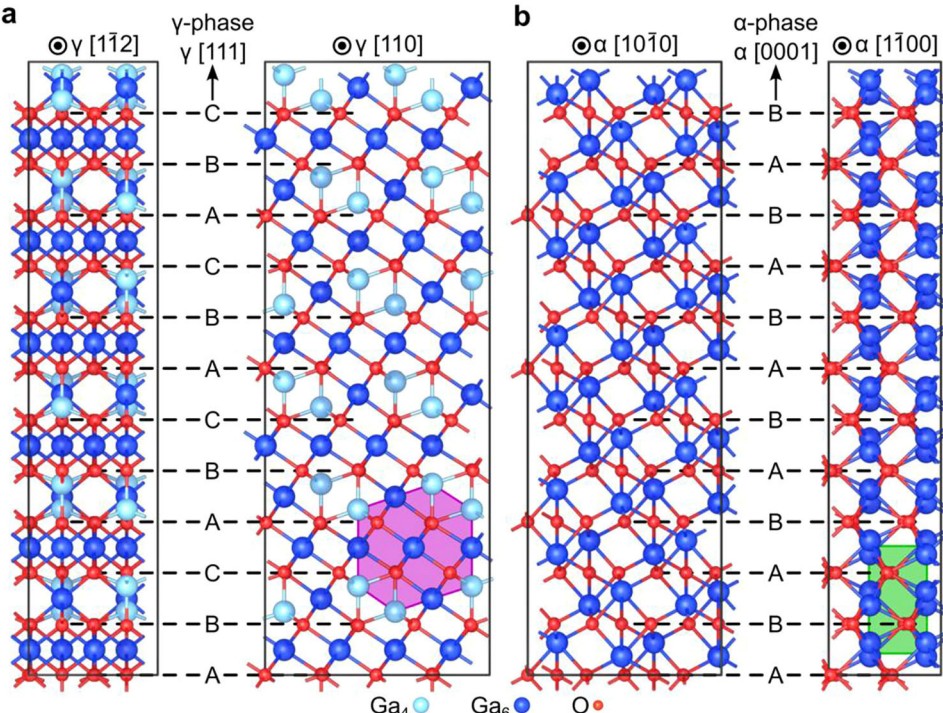

**Fig. 1 | Structural relationship between different Ga2O3 polymorphs.** The orthogonal supercells of (**a**) γ-and (**b**) *α*-Ga$_2$O$_3$ polymorphs. Ga atoms are represented as big blue (color-coded by coordination numbers), and O atoms are small red spheres. Notably, the γ phase adopts a defective spinel structure, where, in a simplified two-Ga-site model, the Ga site occupation is reduced from 1 to maintain the stoichiometric ratio of Ga$_2$O$_3$. The purple hexagon and green rectangle highlight typical Ga patterns observed in the γ and α phases, respectively (refer to Fig. 4).

metastable rhombohedral polymorph (α-Ga$_2$O$_3$, Fig. 1b), there are only a few studies devoted to the radiation defect formation[20–22]; however, indicating that α-phase is more radiation resistant at the range of the nuclear stopping power maximum as compared to that of β-Ga$_2$O$_3$[20]. Concurrently, the disorder buildup in α-Ga$_2$O$_3$ involves surface amorphization, somewhat resembling the features observed in GaN[23,24]. Nevertheless, if disorder-induced polymorphism is realized in α-Ga$_2$O$_3$, its impact on the device applicability may be even more interesting than that in β-Ga$_2$O$_3$; since α-Ga$_2$O$_3$ exhibits the widest bandgap among the rest of the Ga$_2$O$_3$ polymorphs family[25,26], making it more likely to anticipate higher band offsets in, e.g., γ/α interfaces[27].

Thus, in the present work, we undertake a systematic investigation of the radiation phenomena in α-Ga$_2$O$_3$ and determine conditions sufficient for igniting α-to-γ polymorph transition. We argue that in contrast to that in β-phase, the O sublattice in α-phase possesses a hexagonal close-packed (hcp) structure (Fig. 1b) so that to activate α-to-γ phase transformation, significant structural rearrangements are required in both Ga and O sublattices. Moreover, consistent with predictions from the energy diagram, α-to-γ phase transformation requires the accumulation of the substantial tensile strain to initiate otherwise impossible lattice glides. Thus, we explain these fascinating phase transformation trends in terms of the combination of disorder and strain-governed process. As a result, we demonstrate atomically abrupt α/γ interfaces paradoxically self-organized out of the stochastic disorder.

## Results and discussion

Figure 2 provides a survey of the experimental data, including systematic measurements of the samples crystallinity as a function of displacement per atom (dpa) obtained by (a) Rutherford backscattering spectrometry in channeling mode (RBS/C) and (b) x-ray diffraction (XRD) in combination with scanning transmission electron microscopy (STEM) cross-sections of the selected samples in panels

(c)-(f), associated with characteristic process stages as illustrated in cartoon insets in the middle of the figure. Indeed, already for low dpa conditions, i.e., ≤ 4 dpa, the RBS/C spectra reveal - consistently with the literature[20,21] – a surface disorder peak (see Fig. 2a), specifically at dpa = 4 corresponding to a 6 nm thick amorphous layer, as confirmed by TEM data in Fig. 2c. In addition, there is a broader "bulk" disorder peak localized far beyond of the maximum of the primary defect generation ($R_{pd} \approx 105$ nm) according to the SRIM code[28] simulations. This stage is accompanied by a tensile strain accumulation, as clearly seen from the appearance of a shoulder on the left-hand side of the α-Ga$_2$O$_3$ (006) reflection in the XRD 2θ scans (Fig. 2b) for dpa = 4. Further, at the 20 ≤ dpa ≤ 120 range, the surface disorder peak broadens and eventually reaches the random level, while the magnitude of the bulk peak saturates at a much lower disorder level, see Fig. 2a. For example, at dpa = 120, ~130 nm thick amorphous layer is revealed by TEM as illustrated in Fig. 2d. Notably, this disorder accumulation stage does not reveal much of changes in the XRD spectra, except of the strain release, as seen from the evolution of the left-hand side parts of the (006) diffraction peak in Fig. 2b.

Spectacularly, an additional relatively tiny dpa increase - just by a few tens of percent - dramatically changes the structure. Indeed, at dpa = 140, RBS/C intensity increases right behind the surface amorphous layer, see the 100–170 nm range below the surface in Fig. 2a. This prominent transformation is accompanied by the appearance of a new diffraction peak centered at ~37.7°, see Fig. 2b, which is identified as γ-Ga$_2$O$_3$ (222) reflection[29], in agreement with TEM data in Fig. 2e. Further dpa increase improves the crystallinity in this region as clearly seen from the decreased the RBS/C yield at dpa = 160 as compared to that at dpa = 140, see Fig. 2a. Simultaneously, the width of the phase-modified layer expands with increasing dpa. Notably, the γ-layer expands both into the crystal bulk in the form of α-to-γ transition and towards the surface, so that the amorphous layer converts into γ-phase too, as schematically shown in the corresponding cartoon inset. Thus,

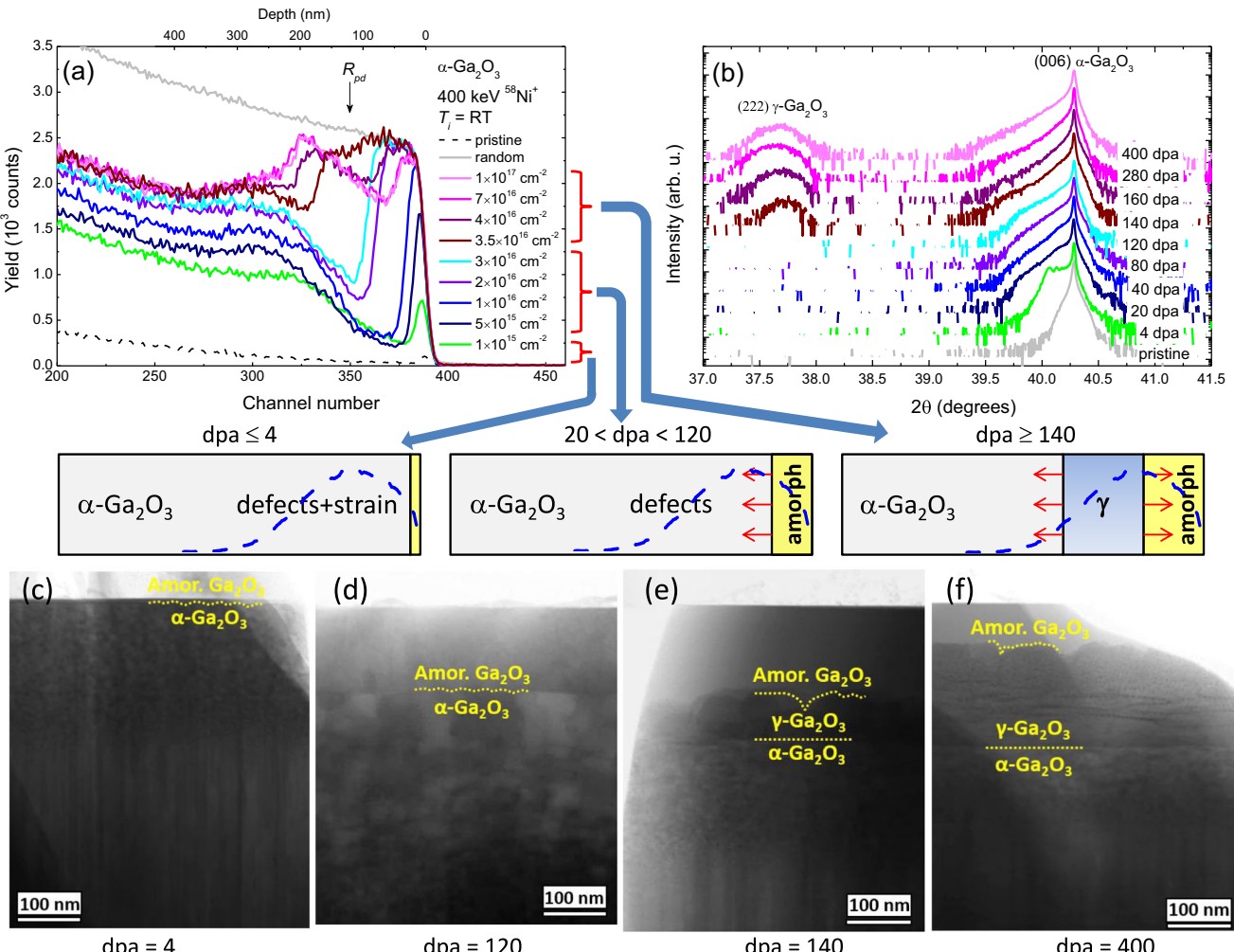

**Fig. 2 | Phase transformations in α-Ga₂O₃ as a function of dpa. a** Rutherford backscattering spectrometry in channeling mode (RBS/C) spectra and (**b**) corresponding x-ray diffraction (XRD) 2θ scans of the α-Ga₂O₃ samples irradiated with 400 keV Ni ions. The doses and corresponding dpa values are indicated in legends in panels (**a**) and (**b**), respectively. The maximum of the primary defect generation ($R_{pd}$) is indicated by the arrow in panel (**a**). Panels (**c**–**f**) show low magnification bright field (BF) scanning transmission electron microscopy (STEM) cross-sections of the selected samples for dpa values highlighting characteristic trends, in correlation with cartoons in the middle of the figure, where the primary defect generation profiles obtained by the SRIM code[28] simulations are shown by the dashed lines. The red arrows in the cartoons indicate the expanding of the amorphous layer and γ-phase with increasing dpa value.

the surface amorphous layer broadens as a function of dpa until the α-to-γ phase transformation starts, while further dpa increase leads to the shrinkage of the amorphous layer due to the γ-film expansion. Notably, dpa increase beyond 140 dpa has practically no impact on the crystallinity of the newly formed γ-phase, confirming its remarkable radiation tolerance consistently with literature[15]. Moreover, comparing it to β-phase, α-Ga₂O₃ itself can be indeed classified as a higher radiation tolerant material, since α-to-γ phase transition starts at much higher dpa levels (N.B. dpa = 1 was shown to be sufficient to start β-to-γ transition[15]). In order to exclude the possible role of the implanted Ni atoms on the observed α-to-γ phase transition, we performed an additional implantation with chemically inert Au ions. According to RBS/C and XRD data shown in Supplementary Note 3 the α-to-γ phase transformation is clearly observed also for Au implantation, indicating that the observed phenomena are related to the accumulated disorder rather than to the chemical effects of the implanted ions.

Meanwhile, another spectacular observation indicated already by the data in Fig. 2e and f is an ultimate abruptness of the γ/α interfaces resulted out of colossal disordering process having a stochastic nature. To investigate this phenomenon in detail, we performed high-angle annular dark field (HAADF) STEM analysis of the interfaces obtained in

this study, see Fig. 3. Specifically, Fig. 3a and b show the high-resolution images of the amorphous/α-phase and amorphous/γ-phase interfaces formed in the samples upon disordering with dpa = 4 and 400, respectively. As expected from the stochastic nature of the disorder, these interfaces are not abrupt, and in the case of the amorphous/γ-phase interface even rather rough. However, even it is contraintuitive, polymorph interfaces formed out of the same stochastic disorder, specifically γ/α interface is atomically sharp as clearly demonstrated by Fig. 3c, d showing high-resolution HAADF STEM images of the interface region taken along different zone axes in the sample subjected to dpa = 400. The corresponding selected area electron diffraction (SAED) patterns for each image and the schematic unit cell and lattice stackings oriented as in the interfaces are shown in the insets and at the right-hand sides of the corresponding images, respectively. Note that both the schematic unit cells and the lattice stacking were directly obtained from the indexing of the electron diffraction patterns shown in the respective figures collected for each phase. Specifically, the orientation relationships at the γ/α interface were determined from STEM as γ[110] / α[1$\bar{1}$00] and γ[112] / α[10$\bar{1}$0] and were used further in simulations to shed more light on the mechanism of the γ/α interface formation.

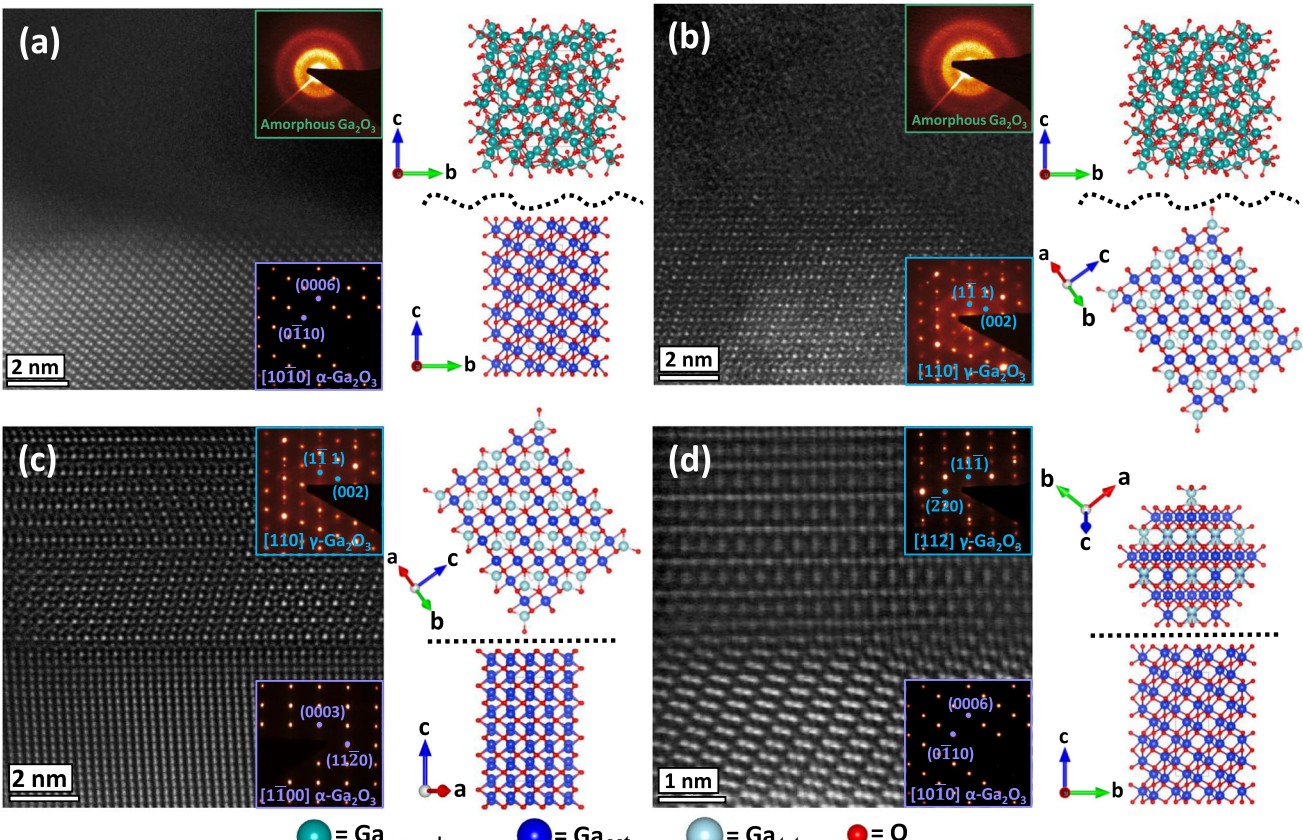

= Ga amorphous        = Ga oct        = Ga tet        = O

**Fig. 3 | Quality of interfaces formed out of stochastic disorder.** Selected area electron diffraction (SAED) patterns and high-angle annular dark field (HAADF) scanning transmission electron microscopy (STEM) images of (**a**) non-abrupt amorphous/α and (**b**) amorphous/γ interfaces in the samples with 4 and 400 dpa, respectively, in contrast to atomically sharp γ/α interface imaged along the (**c**) [110]

γ-Ga₂O₃ / [1100] α-Ga₂O₃ and (**d**) [112] γ-Ga₂O₃ / [10̄10] α-Ga₂O₃ zone axes for the sample with dpa = 400. The projected structure model for each phase is depicted on the right-hand side of the images, where the color code for spheres are turquoise (amorphous Ga), dark blue (octahedral Ga), light blue (tetrahedral Ga), and red (O). The interfaces in the insets are shown by the dashed black lines.

Thus, for that matter, we performed machine-learning-driven molecular dynamics (ML-MD) simulations, and Fig. 4 summarizes the atomic configurations and dynamic evolution of such an interface. The initial local atomic configuration of the ML-MD interface [Fig. 4c] closely resembles the interface observed in high-resolution STEM images [Fig. 4a and b]. The γ-O and α-O sublattices follow fcc (A'B'C'-A'B'C'…) and hcp (AB-AB…) stacking orders, respectively, as indicated in Fig. 4d. Consequently, the initial γ/α interfacial transition region (cyan region in Fig. 4b–d) exhibits a stepped edge with two horizontally mismatched (A'|B) and (C'|A) O stacking layers and a vertically mismatched B'-B stacking order which is energetically unfavorable. The dynamic evolution of the representative (A'|B) O layers (black box in Fig. 4d) is further detailed in Fig. 4e, and Supplementary Movie 1 illustrates the complete simulation. Within the first 5 ps, the α-O B stacking layer reconstructs into a γ-O C'-like stacking, accompanied by overall vertical lattice distortion, leading to a transient local hcp-like stacking. Further plane displacements follow typical hexagonal directions alone [110] or 1̄10, as indicated by the blue arrows in Fig. 4e. These plane displacements or "glides" complete at $t = 355$ ps, along with simultaneously rapid local Ga rearrangement (see Supplementary Movie 1, from frame 3450 to frame 3560, $t = 345 \sim 356$ ps). The final γ/α interface presents a perfectly lattice-aligned O (B' = B) single layer with ultimate atomic sharpness.

Despite that β-to-γ phase transformations also occurs under ion irradiation, the mechanism of the disorder-induced transformations in α-Ga₂O₃ is dramatically different from that in β-phase[15]. Indeed, previously it was demonstrated that β-to-γ phase transformation occurs

due to accumulation of Ga disorder, while O sublattice having fcc structure for both phases exhibits a strong recrystallization trend within collision cascade[15,17]. In contrast, the O sublattice in the α-phase possesses hcp structure, so the structural rearrangements in both sublattices are required for the α-to-γ phase transformation. Furthermore, out of the energy consideration, α-to-γ phase transformation requires an accumulation of the tensile strain in the system, see Supplementary Note 1 and ref. 30. Literally, in order to realize the α-to-γ Ga₂O₃ phase transformation, the hcp α-O sublattice must transform to fcc γ-O sublattice. Both hcp and fcc stackings share efficient closed-packing arrangements, suggesting that the transformation likely occurs via a slip of closed-packed layers. However, the atomic volume differences between the two phases are the biggest among all polymorphs. The smallest atomic volume of the α phase is around 10.1 Å³/atom while that of the γ phase is around 11 Å³/atom, see Supplementary Note 1. This indicates that such phase transformation needs to be assisted with an expansion of the system.

To understand the expansion mechanism of α-to-γ phase transformation, we systematically compare O sublattice parameters of α and γ-Ga₂O₃ phases, as analyzed in Fig. 5. Indeed, Fig. 5a illustrates the stacking of the closed-packed planes perpendicular to the closed-packed layers of the α-O sublattice (AB-AB-AB) and γ-O sublattice (A'B'C' -A'B'C'). These data indicate that the interlayer distances in the α-O sublattice are smaller than those in the γ-O sublattice (see the corresponding levels marked by the black dashed lines in Fig. 5a). Further, the PRDF curves of the single layer in the O sublattice which is perpendicular to the close-packed layers (Fig. 5a), the A-A distances in

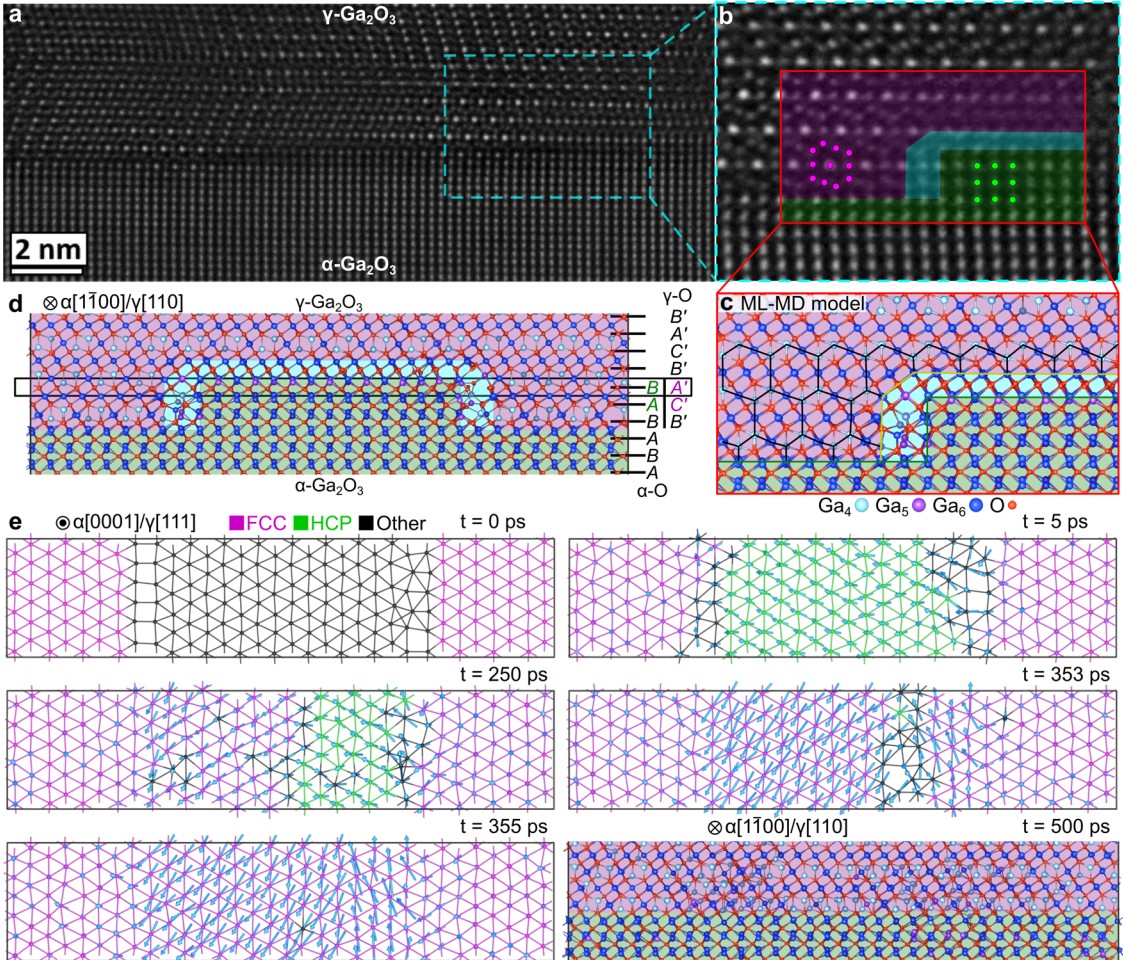

**Fig. 4 | Atomic configuration and dynamic evolution towards atomically sharp γ/α interface. a** Experimentally measured γ/α interface as that in Fig. 3c. **b** Enlarged interface image with the simulation template inset illustrating corresponding initial configurations of the machine-learning-driven molecular dynamics (ML-MD) model in panel (**c, d**) detailed configuration in terms of the O sublattice repetitions in γ-Ga$_2$O$_3$ (γ-O) and α-Ga$_2$O$_3$ (α-O); Ga atoms are big blue (color-coded based on coordination numbers) and O atoms are small red spheres. The γ-phase, boundary, and α-phase regions are colored in purple, cyan, and green, respectively. **e** Dynamic evolution of the initially mismatched oxygen (A′ | B) stacking layer [labeled by the black box in panel (**d**)] at 900 K, top-viewed from α[0001]/γ[111] axis, demonstrating the planar slip displacements relative to the initial positions, as indicated by the blue arrows (note, that displacement magnitudes are scaled by a factor of 3 for clarity). The color-coding of oxygen atoms illustrates local stacking types, such as fcc and hcp. At $t = 500$ ps, the atomically sharp γ/α interface forms with a perfectly matched oxygen (B = B′) stacking layer in accordance with the ML-MD simulation (side-viewed from α [1$\bar{1}$00]/γ[110] axis). See Supplementary Movie 1 for the whole evolution process.

the α-O sublattice is the peak of $D_{A-A}$ at 4.5 Å, the A′-A′ distances in the γ-O sublattice is $D_{A'-A'}$ at 7.1 Å. Accounting these values, the average interlayer distances in the closed-packed layer of α-O and γ-O sublattice is 2.25 Å and 2.37 Å, respectively. This indicates an expansion between the closed-packed layers of the α phase (alongside α[001] orientations) during the phase transformation of ~5%. Within the closed-packed layers, as shown in Supplementary Note 2, the PRDF curves indicate negligible expansion, particularly at longer distances. This implies that even though there are some deviations between the arrangements of efficient packing, no prominent "isotropic" expansion is observed within these closed-packed planes.

Meanwhile, the anisotropic expansion of α-Ga$_2$O$_3$ may play a prominent role in explaining the mechanism of α-to-γ Ga$_2$O$_3$ phase transformation under irradiation. To investigate the expansion evolution, we employ additional isothermal-isobaric ensemble (NPT) relaxations with anisotropic barostat, after every 100 primary knock-on atoms (PKAs) overlapping collision cascade simulations to provide a sufficient degree of freedom for the system volume adjustment. Figure 5c displays the strain in the three coordination directions, revealing significant expansion along the $z$-axis, especially during the

early stages of the radiation defects accumulation. Conversely, the system sizes in the $x$ and $y$-directions show minimal changes before 300 PKAs, followed by a slight expansion. This anisotropic expansion aligns with the lattice differences between α and γ-Ga$_2$O$_3$ phases. The lattice experiences the strongest expansion along the $z$-axis, which is perpendicular to the closed-packed plane on the oxygen sublattice, while the $x$ and $y$ axes belong within the closed-packed plane, and the expansion in these directions is very small. The structure in the left-hand side of Fig. 5d shows the lattice of the pristine α-Ga$_2$O$_3$. One can see that one of each three octahedral sites within the hcp O sublattice is vacant. After collision cascades, the defective α-Ga$_2$O$_3$ (the right-hand side of Fig. 5d) shows that the displaced Ga atoms readily occupy the available octahedral interstitial positions, forming Ga-Ga pairs with incredibly short distances of less than 2.4 Å (visualized in Fig. 5d by blue sticks). The appearance of the multitude of the short-distance Ga-Ga pairs aligned with the $z$-axis in the defective α-Ga$_2$O$_3$ (see Fig. 5e) demonstrates that Ga defects indeed primarily occupy available octahedral sites in the hcp oxygen sublattice. These defects accumulate stress, which relaxes, straining the lattice in the $z$-axis direction, as seen in the experiment. Indeed, after the NPT relaxation with

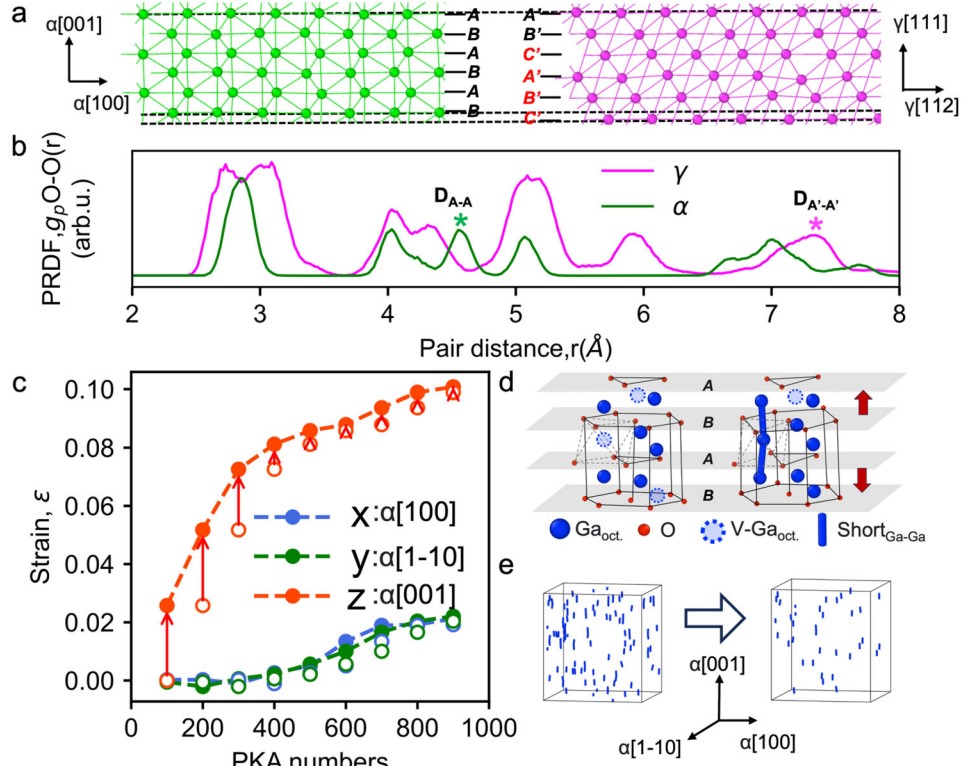

**Fig. 5 | Anisotropic expansion of α-Ga₂O₃ triggered by disorder. a** Oxygen sublattices of $\alpha$-Ga$_2$O$_3$ (left) and $\gamma$-Ga$_2$O$_3$ (right) shown as A-B planes in the hcp stacking for $\alpha$-Ga$_2$O$_3$ and as A′-B′-C′ planes in the fcc stacking for $\gamma$-Ga$_2$O$_3$ stacked perpendicularly to the close-packed direction for both lattices. **b** The corresponding partial radial pair distribution function (PRDF) curves for the oxygen slabs of $\alpha$-Ga$_2$O$_3$ (green) and $\gamma$-Ga$_2$O$_3$ (purple) shown in (**a**). Asterisks highlight the peaks of the A-A and A′-A′ distances in both structures. **c** The expansion fraction of the $\alpha$-Ga$_2$O$_3$ along the $x$, $y$, and $z$-axes during the collision cascade simulations, the $x$, $y$, and $z$-axes correspond to $\alpha$[100] (blue), $\alpha$110 (green), and $\alpha$[001] (red)

orientations, respectively. The hollow and solid circles correspond to the system before and after isothermal-isobaric ensemble (NPT) relaxation with anisotropic barostat, while the dashed lines are to guide the eyes only. **d** Partial enlargement images of the pristine (left) and defective (right) $\alpha$-Ga$_2$O$_3$ during the collision cascade simulations. The Ga and O atoms are large blue and small red spheres, respectively. The short-distance Ga-Ga neighbors (distance shorter than 2.4 Å) are highlighted with the blue sticks. **e** The evolution of the number of short-distance Ga-Ga neighbors (purple sticks) randomly formed in $\alpha$-Ga$_2$O$_3$ after 100 knock-on atoms (PKA) collision cascades from before (left) to after (right) NPT relaxation.

anisotropic barostat, the system expands along the $z$-axis, and the number of Ga-Ga short-distance pairs decreases significantly (see Fig. 5e). In other words, our results suggest that the accumulation of Ga defects generates the stress between the oxygen closed-packed planes, leading to significant expansion. The fully relaxed stress in our simulations within the NPT ensemble, the system expands by approximately 10% in the $z$-axis. This expansion is greater than 5% needed for the transformation of the hcp $\alpha$-Ga$_2$O$_3$ oxygen sublattice into the fcc $\gamma$-Ga$_2$O$_3$ oxygen sublattice according to the mechanism proposed in Fig. 3. A full relaxation of the accumulated stress can be expected only near the surface. The large expansion of the lattice allows for easier accumulation of the defects, which leads to faster deterioration of the crystal lattice. This is why the lattice of $\alpha$-Ga$_2$O$_3$ near the surface does not transform into the stable $\gamma$-Ga$_2$O$_3$ phase under ion irradiation, but first becomes amorphous, see Fig. 2. However, in deeper regions, stress generated by octahedral Ga interstitials is not easily released, controlling interplane distances to be closer to the $\gamma$-O sublattice. As stress increases, crystal plane slip becomes highly likely, completing the phase transformation from $\alpha$ to $\gamma$ deep beneath the surface. Finally, it should be noted that despite the surface amorphous layer potentially affecting the $\alpha$-to-$\gamma$ phase transformation, the prime condition required for the phase transformation is a sufficiently large anisotropic strain accumulated due to the radiation disorder. This is also supported by higher energy implants, as shown in Supplementary Note 4, where the phase transformation region is clearly separated from the surface.

In conclusion, disorder-induced ordering and unprecedently high radiation tolerance in the $\gamma$-phase of gallium oxide is a recent spectacular discovery at the intersection of fundamental physics and electronic applications. Importantly, before the present work, all these amazing literature data were collected with initial samples in the form of the thermodynamically stable $\beta$-phase of this material. Here, we investigated these phenomena starting instead from the already metastable $\alpha$-phase and explained a radically new trend occurring in the system. We argued that in contrast to that in $\beta$-to-$\gamma$ disorder-induced transitions, the O sublattice in $\alpha$-phase exhibits hexagonal close-packed structure, so that to activate $\alpha$-to-$\gamma$ transformation, significant structural rearrangements are required in both Ga and O sublattices. Moreover, consistently with theoretical predictions, $\alpha$-to-$\gamma$ phase transformation requires accumulation of the substantial tensile strain to initiate otherwise impossible lattice glides. Thus, we explain the experimentally observed trends in terms of the combination of disorder and strain governing the process. Finally, and perhaps most amazingly, we demonstrate atomically abrupt $\alpha/\gamma$ interfaces paradoxically self-organized out of the stochastic disorder. As such, the present data broadens the research community interest in Ga$_2$O$_3$ polymorphs and paves the way for the generalization of such polymorphism phenomena toward other materials.

## Methods

In the present work, we used rhombohedral ~1 μm thick $\alpha$-Ga$_2$O$_3$ films grown on sapphire substrates by halide vapor phase epitaxy (see

details of the synthesis elsewhere[31]). The samples were implanted at room temperature with 400 keV $^{58}$Ni$^{+}$ ions in a wide dose range ($1 \times 10^{15}$ – $1 \times 10^{17}$ Ni/cm$^2$), keeping the ion flux constant at $6 \times 10^{12}$ at.cm$^{-2}$s$^{-1}$. All the implants were performed at 7° off-angle orientation from a normal direction to minimize channeling. Simulations with Stopping and Range of Ions in Matter (SRIM) code[28] were used to get the primary defect generation profile and calculate dpa values. This computer program simulates the interaction between energetic ions and target materials using a binary collision approximation. According to the simulations, the depth where the defect generation is maximal, which corresponds to the maximum of the nuclear energy deposition, is equal to $R_{pd} = 105$ nm for Ni implant parameters used in the present study. The dpa values were calculated for each ion dose using conventional methodology[32]. Specifically, the quoted dpa values were taken at the maximum of the SRIM vacancy generation profiles simulated for a given ion dose and normalized to the atomic density of α-Ga$_2$O$_3$ ($n_{at} = 10.35 \times 10^{22}$ at/cm$^3$). The SRIM simulations were performed in a full damage cascade mode with 28 eV and 14 eV as the displacement energies for Ga and O atoms, respectively[33]. Two control samples were implanted at room temperature with 1.2 MeV Au ions to a dose of $1 \times 10^{16}$ cm$^{-2}$ and 800 keV Ni ions to a dose of $5 \times 10^{16}$ cm$^{-2}$.

Structural characterization of the implanted samples was performed by a combination of RBS/C, XRD and STEM. The RBS/C measurements were performed by 1.6 MeV He$^{+}$ ions incident along [001] direction in the α-Ga$_2$O$_3$ part of the structure and backscattered into a detector placed at 165° relative to the incident beam direction. XRD 2θ measurements were performed using the RIGAKU SmartLab diffractometer with high-resolution Cu K$_{\alpha 1}$ radiation and Ge(440) four-bounced monochromator. For cross-sectional STEM studies, selected samples were thinned by mechanical polishing and by Ar ion milling in a Gatan PIPS II (Model 695), followed by plasma cleaning (Fishione Model 1020) immediately before loading the samples into a Cs-corrected Thermo Fisher Scientific Titan G2 60–300 kV microscope, operated at 300 kV. The STEM images were recorded using a probe convergence semi-angle of 23 mrad, a nominal camera length of 60 mm using two different detectors: HAADF (collection angles 100–200 mrad), and BF (collection angles 0–22 mrad). The structural model of different phases was displayed using VESTA software[34].

The ML-MD simulations were conducted using LAMMPS package[35]. The self-developed ML interatomic potential of the Ga$_2$O$_3$ system was employed[30], which was designed with high accuracy for all five experimentally known Ga$_2$O$_3$ polymorphs and generality for disordered structures. Furthermore, the short-range repulsive interactions are explicitly fitted to high-accuracy ab initio data, and their reliability has been extensively tested in our previous studies[15,17–19]. The evolution of γ/α interface is simulated using an orthogonal cell comprising 15,360 atoms with side lengths of ~ 82.2 × 17.8 × 113.4 Å$^3$. The *x*, *y*, and *z*-axes correspond to γ[112]/α[100], γ[110]/α1$\bar{1}$0, and γ[111]/α[001] orientations, respectively. The cell is initially optimized to a local minimum and is further run at 900 K and 0 bar in NPT ensemble using the Nosé-Hoover algorithm[36] for 500 ps with 1 fs per MD step. The polyhedral template matching (PTM) method[37] is employed to identify the local stacking structure of the O sublattice. The coordination number of Ga atoms are counted with the cutoff radius of 2.6 Å. The structural analyses and visualization are done with OVITO software[38].

A total of 900 overlapping cascades were conducted on the α-Ga$_2$O$_3$ cell, which contains 14,400 atoms in a ~51 × 53 × 55 Å$^3$ box. This scale of the simulation cell was taken to prevent cascade overlapping the temperature-controlled borders and to minimize computational time. In each iteration, the Ga or O atom was randomly selected as the PKA. The PKA was assigned a kinetic energy of 500 eV with a uniformly random momentum direction. To maintain consistency, the entire cell was translated and wrapped at periodic boundaries, positioning the PKA at the center of the cell. Each simulation iteration consisted of two periods. During the first cascade period, the cell was thermalized using NVE-MD for 5000 MD steps with an adaptive time step. Electron-stopping frictional forces were applied to atoms with kinetic energies above 10 eV[39,40]. In the subsequent period, the simulations continued in a quasi-canonical ensemble with a Langevin thermostat[41] applied to border atoms (within 7.5 Å of the simulation box boundaries, redefined in each iteration) for 10 ps at 300 K, to mimic the heat dissipation from the cascade area into the bulk materials. In addition, relaxation simulations were periodically conducted after every 100 cascades. During these relaxation simulations, the system was subjected to NPT ensemble at 300 K and 0 bar for 100 ps, with temperature controlled by the Nosé-Hoover thermostat and pressure controlled by the anisotropic barostat[36].

## Data availability
The data that support the findings of this study are available within the paper and its Supplementary Information file. The machine-learning potential parameter files used to run classical MD simulations are openly available at https://doi.org/10.6084/m9.figshare.21731426. The corresponding raw data of the classical MD of γ/α interface published in this paper are openly available at https://doi.org/10.6084/m9.figshare.28236944. Source data are provided in this paper.

## Code availability
The code and software used in this work are LAMMPS, VASP, OVITO, and SRIM, which are openly available online from the corresponding developers and maintainers.

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

## Acknowledgements

M-ERA.NET Program is acknowledged for financial support via the GOFIB project (administrated by the Research Council of Norway project number 337627 in Norway and the Academy of Finland project number 352518 in Finland). Additional support was received from the DIOGO project funded by the Research Council of Norway in the frame of the FRIPRO Program project number 351033. The experimental infrastructures were provided at the Norwegian Micro- and Nano-Fabrication Facility, NorFab, supported by the Research Council of Norway project number 295864, at the Norwegian Center for Transmission Electron Microscopy, NORTEM, supported by the Research Council of Norway project number 197405. J.Z. acknowledges the National Natural Science Foundation of China under Grant 62304097, Guangdong Basic and Applied Basic Research Foundation under Grant 2023A1515012048; Shenzhen Fundamental Research Program under Grants JCYJ20230807093609019 and JCYJ20240813094508011. Computing resources were provided by the Finnish IT Center for Science (CSC) and by the Center for Computational Science and Engineering at the Southern University of Science and Technology. The paper was also supported by the "Strategic R&D program" funded by the Korea Institute of Ceramic Engineering and Technology (KICET), Republic of Korea, in 2024 (KPP23004-0-02) The international collaboration was also fertilized via INTPART Program at the Research Council of Norway project number 322382 as well as UTFORSK Program at the Norwegian Directorate for Higher Education and Skills project number UTF–2021/10210.

## Author contributions

A.K. and A.A. conceived the research strategy and designed the methodological complementarities. J.H.P. and D.W.J. contributed to the crystal growth and provided the samples. A.A. and J.G.F. carried out experiments and provided initial drafts for the description of the experimental data. R.H. and J.Z. performed molecular dynamics simulations. F.D., R.H., and J.Z. developed the theoretical models and composed the theoretical part of the manuscript. A.K. and A.A. finalized the manuscript with the input from all the co-authors. All co-authors discussed the results as well as reviewed and approved the manuscript. A.K., Ø.P., J.Z., and F.D. administrated their parts of the project and contributed to the funding acquisition. A.K. coordinated the work of the partners.

## Competing interests

The authors declare no competing interests.
