## [Transparent Peer Review file · Nature Communications]

Phase glides and self-organization of atomically abrupt interfaces out of stochastic disorder in α -Ga₂O₃

Corresponding Author: Dr Alexander Azarov

Version 0:

Reviewer comments:

Reviewer #1

(Remarks to the Author)

This paper reports on an experimental and computational investigation of the effects of radiation on the alpha phase of Ga₂O₃ and on the resulting alpha-to-gamma polymorph transition. Ga₂O₃ is a wide-gap semiconductor with potential applications in e.g. power electronics and photodetectors, as well as radiation-tolerant devices for space applications.

The authors investigate the structural rearrangements required to activate the alpha-to-gamma phase transformations. They show that such rearrangements are driven by the combination of disorder and strain and involve both the Ga and O sublattices, in contrast to the case of beta-to-gamma transformations.

As far as I can say, experiments and simulations have been carried out carefully and the methodology is sound. The results are definitely interesting, however the authors should address the following points before I can recommend publication of the paper in Nat. Comm.

The manuscript is written in a style suitable to a specialized journal, not to a broad-audience journal such as Nature Communications. The paper is a difficult read for non-experts. The authors should devote some extra efforts to improve the clarity. More specifically, I recommend the authors to make the following changes:

- It would be useful to add a figure in the introductory part (or at the beginning of the results section) illustrating the structure of the alpha, beta and gamma phases. This would also help the reader understand the insets of Figure 2 (see also the point below) and Figure 3.
- Figure 1: I would use a different color scheme in plots a and b to improve readability. For instance, one could use different colors for the 3 groups of curves corresponding to $dpa \leq 4$, $20 < dpa < 120$ and $dpa > 140$, respectively, and different shades of the same color within each group.
- At page 4, it is mentioned that "In addition there is a broader "bulk" disorder peak localized far beyond of the maximum of the primary defect generation ($R_{pd} \approx 105$ nm) according to the SRIM code [28] simulations". The authors should explain what the SRIM code simulations consist of. There is a brief discussion of the SRIM simulations in the Methods section but, in my opinion, it does not clarify the meaning of the sentence above to non-experts.
- How were the "schematic unit cell and lattice stackings" in Figure 2 exactly obtained?
- Figure 3 is difficult to understand: first of all, the grey shadow in panel (d) is hardly visible. Furthermore, the color schemes are misleading: the authors changed the color coding for the Ga atoms with respect to Figure 2 (they were green in Fig. 2 and now they are blue). More importantly, since the authors used green and purple colors for the alpha and gamma phase in panel (d), it would be better to use green and purple colors for the HCP (alpha-phase) and FCC (gamma) stacking, not red and green. Also it should be explained more clearly that the first 5 plots of panel (e) correspond to a different view with respect to the 6th plot (and to the panel (d)), if I understood correctly. One should add proper axes in the figures to explain from which axis the view is taken.
- Figure 4: Similarly to Figure 3, I would change the color scheme in (a) and (b): green for alpha and the HCP stacking,

purple for gamma and the FCC stacking. (d-e) The authors keep changing the color scheme for the atomic species: now Ga atoms (and sticks connecting them) are purple. Furthermore, the red arrows depicting the displacement vectors (of the O atoms?) are hardly visible. Figure (e) is not particularly useful: I would move it to the supplement and increase its size to make it more understandable.

- Additional explanations about the overlapping collision cascade simulations should be provided. Why did the authors perform quasi-canonical simulations (with a Langevin thermostat applied to border atoms) after the NVE run and before the NPT run?

- An accurate description of short-range interactions is crucial for ion irradiation simulations. This is not easy to achieve with standard GAP machine-learning potentials. This point is not mentioned at all in the manuscript and I had to read Ref. 30 to find out that the short-distance accuracy of the potential was improved by employing a set of external repulsive pair potentials fitted to the ab initio data and by adding high-energy structures in the DFT database. This is an important point, which, in my opinion, deserves to be mentioned in the Methods section to convince the reader that the computational approach employed is sensible.

- All acronyms should be properly defined at their first occurrence, not in the Methods section.

- The conclusion section is sort of disappointing. It briefly summarizes the main results but doesn't put them in a broader context. Are the results relevant to other materials? Is the alpha-phase of Ga₂O₃ promising for applications? This is briefly mentioned in the introductory part (by the way, the authors cite a paper under preparation, Ref. [26]: it would be better to provide explicitly the values of the bandgaps of the phases instead of referring to a paper which is not available) but I would have expected the discussion to be expanded in the conclusions to address advantages and disadvantages of the alpha phase compared to the stable beta phase.

Typos and minor points:

- pag. 4 "peak broadens and eventually reaching the random level" -> "peak broadens and eventually reaches the random level"

- pag. 7 "posses" -> "possesses"

- pag. 11: "The evolution of the gamma/alpha interface are stimulated" -> "The evolution of the gamma/alpha interface is simulated"

- pag. 12 and caption of Figure 4: replace "isothermal-anisobaric" with "isothermal-isobaric"

- I would refrain from using terms like "spectacular" (used 4 times), "incredibly", "amazing" in a scientific paper but I guess this is a matter of taste

Reviewer #2

(Remarks to the Author)

The authors present experimental evidence of the irradiation-induced formation of the γ phase, which has a very high irradiation tolerance, in metastable α -Ga₂O₃. This process is shown by molecular dynamics simulations to involve O lattice glide, which is absent in γ phase formation in stable β -Ga₂O₃. The γ - α interface formed out of disorder is very sharp as shown by HRTEM measurements. In my opinion, this new pathway of disorder-induced ordering in Ga₂O₃ is sound. But to me, there are still some issues in the manuscript that need to be clarified before my recommendation of publication in Nature Communications.

Main issues:

1. The illustration of the mechanism of the α - γ phase transition under irradiation, i.e., why there is a critical thickness of the amorphous region for the α - γ phase transition, does not seem convincing to me. The authors stated: "A full relaxation of the accumulated stress can be expected only near the surface. The large expansion of the lattice allows for easier accumulation of the defects, which leads to faster deterioration of the crystal lattice. This is why the lattice of α -Ga₂O₃ near the surface does not transform into the stable γ -Ga₂O₃ phase under ion irradiation, but first becomes amorphous." But I do not see the reason why the γ phase would not form at the surface with the proper amount of irradiation. The authors' argument needs some quantitative justification. Since the authors already employed molecular dynamics to simulate how the α / γ interface evolves, is it possible to use molecular dynamics to simulate how the γ phase would nucleate in the amorphous/ α heterostructure as the amorphous region thickens?

2. I do not understand why the γ phase grows into the amorphous region as the irradiation increases. Does this suggest that the γ phase is more stable than the amorphous phase under high irradiation?

3. The formation of a sharp α / γ interface, even out of disorder, is not surprising to me. The α -phase side is always ordered, and after the γ phase forms at the surface of the α phase, the interface between these two crystalline phases should be atomically sharp. The interface will be straight to minimize the interfacial energy. Maybe I missed something; the authors can

explain more if that is the case.

4. Would the Ni⁺ irradiation dope any Ni in Ga₂O₃? If so, what will be the role of the doped Ni impurities in the observed γ phase formation and evolution?

Some minor issues:

1. It might be better to show schematics of the unit cells of the α and γ phases at the beginning. This could give readers a clear concept of what the two phases look like at the beginning.

1. In Fig. 1(a), the coloring of the different lines is somewhat hard to track. Consider using colors from cool to warm to label the data from low to high irradiations. In Fig. 1(b), some of the dpa numbers are not easy to recognize. Consider a better way to annotate these numbers. Also consider using larger fonts in Fig. 1(a, b).

2. In Fig. 3(e), the colors for "hcp" and "other" are hard to distinguish. Consider using a more contrasted color scheme.

Version 1:

Reviewer comments:

Reviewer #1

(Remarks to the Author)

The authors have addressed all of my comments and concerns in a satisfactory way. I recommend publication of the manuscript in its present form.

Reviewer #2

(Remarks to the Author)

The authors addressed all my concerns. I now recommend the publication of this manuscript in Nature Communications.

Response on the comments of the Reviewer #1:

Comment 1: “- It would be useful to add a figure in the introductory part (or at the beginning of the results section) illustrating the structure of the alpha, beta and gamma phases. This would also help the reader understand the insets of Figure 2 (see also the point below) and Figure 3.”

Reply: We thank Reviewer #1 for this suggestion. Indeed, the structural complexity of Ga_2O_3 polymorphs presents a major challenge in understanding its polymorphic heterostructures. The β , α , and γ phases exhibit distinct symmetries, monoclinic ($Cm2$), rhombohedral ($R-3c$), and cubic ($Fd-3m$), respectively. As a result, even showing the unit cells alone may not sufficiently useful for clarifying the structural relationships between different polymorphs within a heterointerface system, as shown in Figure R1.

Figure R1. The unit cells of Ga_2O_3 polymorphs as shown in (a) β , (b) α , and (c) γ phases. Notably, the γ phase adopts a defective spinel structure, where, in a two-Ga-site model, the occupation of the six Ga sites is 16/18 to maintain the stoichiometric ratio of Ga_2O_3 .

Therefore, instead of showing the unit cells, the orthogonal supercells of α - and γ - Ga_2O_3 polymorphs, constructed from conventional cells, are shown to illustrate the relevant O-sublattice stacking. As matter of fact, monoclinic β phase cannot be reconfigured in orthogonal supercell with a vertical β $[-201]$ orientation, without breaking the periodic boundary condition in the z direction.

To reply to the similar comment 5 from Reviewer #2. We have added one figure (new Figure 1) in the *Introduction* section with corresponding figure references:

Fig. 1 The orthogonal supercells of (a) γ - and (b) α - Ga_2O_3 polymorphs. Ga atoms are represented as big blue (color-coded by coordination numbers), and O atoms are small red spheres. Notably, the γ phase adopts a defective spinel structure, where, in a simplified two-Ga-site model, the Ga site occupation is reduced from 1 to maintain the stoichiometric ratio of Ga_2O_3 . The purple hexagon and green rectangle highlight typical Ga patterns observed in the γ and α phases, respectively (refer to later Fig. 4).

Comment: 2): “- Figure 1: I would use a different color scheme in plots a and b to improve readability. For instance, one could use different colors for the 3 groups of curves corresponding to $dpa \leq 4$, $20 < dpa < 120$ and $dpa > 140$, respectively, and different shades of the same color within each group.”

Reply: thank you, in the revised ms we have changed the color schemes as suggested the Reviewer. Note that the similar comment was raised also by the second Reviewer.

Comment: 3): “At page 4, it is mentioned that “In addition there is a broader “bulk” disorder peak localized far beyond of the maximum of the primary defect generation ($Rpd \approx 105$ nm) according to the SRIM code [28] simulations”. The authors should explain what the SRIM code simulations consist of. There is a brief discussion of the SRIM simulations in the Methods section but, in my opinion, it does not clarify the meaning of the sentence above to non-experts.”

Reply: In the revised ms we have added more details of the SRIM code in the Methods section.

Comment: 4): “- How were the “schematic unit cell and lattice stackings” in Figure 2 exactly obtained?”

Reply: Both the schematic unit cells and the lattice stacking were directly obtained from the indexing of the electron diffraction patterns shown in the respective figures collected for each phase. Through the analysis and indexing of the diffraction diagrams, we can determine the interplanar distances and angles, thus obtaining the orientation and stacking sequence of each phase in the film. In the revised ms we have added this clarification on p. 6. Furthermore, the color scheme of Fig. 3 (figure 2 of the original ms) has been modified to keep the same color coding throughout the all figures of the ms.

Comment: 5): “- Figure 3 is difficult to understand: first of all, the grey shadow in panel (d) is hardly visible. Furthermore, the color schemes are misleading: the authors changed the color coding for the Ga atoms with respect to Figure 2 (they were green in Fig. 2 and now they are blue). More importantly, since the authors used green and purple colors for the alpha and gamma phase in panel (d), it would be better to use green and purple colors for the HCP (alpha-phase) and FCC (gamma) stacking, not red and green. Also it should be explained more clearly that the first 5 plots of panel (e) correspond to a different view with respect to the 6th plot (and to the panel (d)), if I understood correctly. One should add proper axes in the figures to explain from which axis the view is taken.”

Reply: According to the Reviewer’s suggestion, we have revised Figure 4 (originally Figure 3) Specifically, the additional color coding for Ga atoms is intended to provide clearer distinctions between the γ -Ga₂O₃ (with an ordered Ga₄ and Ga₆ pattern), boundary (with defected Ga₅ sites), and α -Ga₂O₃ (with an ordered Ga₆ pattern) regions, therefore, we have retained this color-coding scheme. We have further updated the colors for the HCP (α -phase) and FCC (γ -phase) stacking, and added proper axes in the figure and caption to better illustrate the viewpoint.

Comment: 6): “- Figure 4: Similarly to Figure 3, I would change the color scheme in (a) and (b): green for alpha and the HCP stacking, purple for gamma and the FCC stacking. (d-e) The authors keep changing the color scheme for the atomic species: now Ga atoms (and sticks connecting them) are purple. Furthermore, the red arrows depicting the displacement vectors (of the O atoms?) are hardly visible. Figure (e) is not particularly useful: I would move it to the supplement and increase its size to make it more understandable.

Reply: We thank Reviewer #1 for the helpful comments. We have adjusted the color scheme in Figure 5 (previously Figure 4) as suggested by the Reviewer, with HCP now in green and FCC in purple. Additionally, the Ga atoms and corresponding sticks have been changed to blue to maintain consistency across the figures.

However, we have decided to keep panel (e) in the main figure, as it is important for demonstrating the significant reduction in Ga-Ga short-distance pairs following the release of lattice strain after the NPT simulations. However, to improve clarity and visibility, we have removed the displacement vectors in panel (e), as this information is already conveyed by the expansion in the z direction.

Comment: 7): “- Additional explanations about the overlapping collision cascade simulations should be provided. Why did the authors perform quasi-canonical simulations (with a Langevin thermostat applied to border atoms) after the NVE run and before the NPT run?”

Reply: Thank you for your comment. The use of a canonical ensemble on the border atoms is intended to control the temperature along the simulation cell boundaries, effectively mimicking heat dissipation as seen in bulk materials. We have updated the explanation in the Methods section of the manuscript to clarify this.

” ... In the subsequent period, the simulations continued in a quasi-canonical ensemble with a Langevin thermostat [41] applied to border atoms (within 7.5 Å of the simulation box boundaries, redefined in each iteration) for 10 ps at 300 K, to mimic the heat dissipation from the cascade area into the bulk material. ...”

Comment: 8): - An accurate description of short-range interactions is crucial for ion irradiation simulations. This is not easy to achieve with standard GAP machine-learning potentials. This point is not mentioned at all in the manuscript and I had to read Ref. 30 to find out that the short-distance accuracy of the potential was improved by employing a set of external repulsive pair potentials fitted to the ab initio data and by adding high-energy structures in the DFT database. This is an important point, which, in my opinion, deserves to be mentioned in the Methods section to convince the reader that the computational approach employed is sensible. “”.

Reply: We thank Reviewer #1 for this great suggestion. Indeed, this essential technical refinement of the GAP potentials forms the foundation for the reliability of our ML-MD ion-irradiation simulations. We have now added the following sentences to the Methods section:

“... Furthermore, the short-range repulsive interactions are explicitly fitted to high-accuracy ab initio data, and their reliability has been extensively tested in our previous studies [15, 17-19]. ...”

Comment: 9): “- All acronyms should be properly defined at their first occurrence, not in the Methods section.”

Reply: Thank you. In the revised ms we have corrected these inconsistencies.

Comment: 10): “- The conclusion section is sort of disappointing. It briefly summarizes the main results but doesn't put them in a broader context. Are the results relevant to other materials? Is the alpha-phase of Ga₂O₃ promising for applications? This is briefly mentioned in the introductory part (by the way, the authors cite a paper under preparation, Ref. [26]: it would be better to provide explicitly the values of the bandgaps of the phases instead of referring to a paper which is not available) but I would have expected the discussion to be expanded in the conclusions to address advantages and disadvantages of the alpha phase compared to the stable beta phase.”

Reply: To start with answering to this comment, Ref. 26 has been corrected, in terms of referring to the published data now. Further, we agree with the added value of setting these results in a broader context and add the corresponding extension to the conclusions: “**As such, the present data broadens the research community interest in Ga₂O₃ polymorphs and paves the way for generalization of such polymorphism phenomena towards other materials.**” Meanwhile, we are not fully certain whether the other part of the ideas mentioned by Reviewer #1 in this comment are readily applicable to elaborate in the conclusion section. Indeed, on one hand, at no doubt, the alpha-phase is higher radiation tolerant than the beta-phase; however, on the other hand, the prime goal of these data was not to compare alpha versus beta phases, but instead to demonstrate the difference in the beta-to-gamma and alpha-to-gamma transformations and corresponding mechanisms. In any case, the most radiation tolerant material of this polymorph family remains to be the gamma-phase as additively shown in this work, supplementing the data in Ref.15.

Comment: 11): “Typos and minor points:

- pag. 4 "peak broadens and eventually reaching the random level" -> "peak broadens and eventually reaches the random level"

- pag. 7 "posses" -> "possesses"

- pag. 11: "The evolution of the gamma/alpha interface are stimulated" -> "The evolution of the gamma/alpha interface is simulated"

- pag. 12 and caption of Figure 4: replace "isothermal-anisobaric" with "isothermal-isobaric"

- I would refrain from using terms like "spectacular" (used 4 times), "incredibly", "amazing" in a scientific paper but I guess this is a matter of taste”

Reply: Thank you. In the revised ms. we have corrected all the typos and made a limited number of corrections to improve the clarity of the ms.

Regarding the use of "isothermal-anisobaric" on page 12 and in the caption of Figure 4 (Fig. 5 in the revised ms), this is not a typo. Since α -Ga₂O₃ is an anisotropic material, we used the isothermal-anisobaric ensemble to study its anisotropic swelling behavior under irradiation. However,

understanding that this terminology is not common and may lead to some misunderstanding, we replace it with the term used in the LAMMPS package as “anisotropic barostat”.

Response on the comments of the Reviewer #2:

Comment: 1) *“1. The illustration of the mechanism of the α - γ phase transition under irradiation, i.e., why there is a critical thickness of the amorphous region for the α - γ phase transition, does not seem convincing to me. The authors stated: “A full relaxation of the accumulated stress can be expected only near the surface. The large expansion of the lattice allows for easier accumulation of the defects, which leads to faster deterioration of the crystal lattice. This is why the lattice of α -Ga₂O₃ near the surface does not transform into the stable γ -Ga₂O₃ phase under ion irradiation, but first becomes amorphous.” But I do not see the reason why the γ phase would not form at the surface with the proper amount of irradiation. The authors’ argument needs some quantitative justification. Since the authors already employed molecular dynamics to simulate how the α/γ interface evolves, is it possible to use molecular dynamics to simulate how the γ phase would nucleate in the amorphous/ α heterostructure as the amorphous region thickens?”*

Reply: It seems that there is some misunderstanding probably related to not perfectly clear discussion of the experimental part of the paper in correlation to the performed simulation. Indeed, we do not claim in the paper that a certain threshold thickness of the amorphous layer is needed for the α -to- γ phase transformation. Instead, the only thing required for the phase transformation is a sufficiently large anisotropic strain accumulated due to radiation disorder. In order to prove that we performed additional experiments, irradiating the same samples with twice-bigger ion energy than that used in Fig. 2. These new data are included into the updated manuscript as Supplementary note 4. In brief, as demonstrated in Supplementary note 4, we have distinctly separated the phase transformation region from the surface where the amorphization may take place. In its turn, our simulations demonstrate that the strain relaxation can easily occur near the surface due to the amorphous layer formation. As a result, the γ phase formation can be suppressed in the near surface region due to surface amorphization and, therefore, strain relaxation in this region. We have added this clarification in p. 10 of the revised ms.

Comment: 2) *“I do not understand why the γ phase grows into the amorphous region as the irradiation increases. Does this suggest that the γ phase is more stable than the amorphous phase under high irradiation?”*

Reply: We agree that this is an interesting phenomenon. Indeed, according to the energy diagram (Suppl. Note 1) the amorphous phase has a higher energy as compared to both α and γ phases, so that the α /amorphous and γ /amorphous interfaces exhibit different stabilities under ion irradiation.

Comment: 3) *“The formation of a sharp α/γ interface, even out of disorder, is not surprising to me. The α -phase side is always ordered, and after the γ phase forms at the surface of the α phase, the interface between these two crystalline phases should be atomically sharp. The interface will be*

straight to minimize the interfacial energy. Maybe I missed something; the authors can explain more if that is the case.”

Reply: We agree with the Reviewer that due to minimization of the interfacial energy the α/γ interface will tend to be atomically sharp. However, the beauty of the phenomenon is that the γ -phase forms due to colossal disordering process having stochastic nature as pointed out in p.5.

Comment: 4) *“Would the Ni+ irradiation dope any Ni in Ga₂O₃? If so, what will be the role of the doped Ni impurities in the observed γ phase formation and evolution?”*

Reply: Indeed, we do not consider the role of the implanted Ni atoms on the observed phenomena. This is related to the fact that both the surface amorphization and phase transformation are related to the disorder accumulation rather than to the chemical effects of the implanted impurities. To support our conclusion we performed additional implantation with Au ion for which any chemical effects on the above mentioned effects should be eliminated. Thus, we added new section in Suppl. Inf. (Suppl. Note 3) and corresponding changes in p. 5 of the revised ms.

Comment: 5) *“Some minor issues:*

1. It might be better to show schematics of the unit cells of the α and γ phases at the beginning. This could give readers a clear concept of what the two phases look like at the beginning.

2. In Fig. 1(a), the coloring of the different lines is somewhat hard to track. Consider using colors from cool to warm to label the data from low to high irradiations. In Fig. 1(b), some of the dpa numbers are not easy to recognize. Consider a better way to annotate these numbers. Also consider using larger fonts in Fig. 1(a, b).

3. In Fig. 3(e), the colors for "hcp" and "other" are hard to distinguish. Consider using a more contrasted color scheme.”

Reply: We have now added new Fig. 1 and revised Figs. 2-5 (originally Figs. 1-4) accordingly. Please refer to the reply to the comments 1 and 4-6 from Reviewer #1.